# The Impact of Social Media and Socio-Cultural Attitudes toward Body Image on the Risk of Orthorexia among Female Football Players of Different Nationalities

**DOI:** 10.3390/nu16183199

**Published:** 2024-09-21

**Authors:** Wiktoria Staśkiewicz-Bartecka, Kommi Kalpana, Samet Aktaş, Gulshan Lal Khanna, Grzegorz Zydek, Marek Kardas, Małgorzata Magdalena Michalczyk

**Affiliations:** 1Department of Food Technology and Quality Assessment, School of Public Health in Bytom, Medical University of Silesia in Katowice, ul. Jordana 19, 41-808 Zabrze, Poland; mkardas@sum.edu.pl; 2Department of Nutrition and Dietetics, School of Allied Health Sciences, Manav Rachna International Institute of Research and Studies, Faridabad 121001, India; kommikalpana80@gmail.com (K.K.); glkhanna56@gmail.com (G.L.K.); 3Department of Training Education, Faculty of Sports Science, Batman University, Batman 72000, Turkey; samet.aktas@batman.edu.tr; 4Department of Sport Nutrition, Jerzy Kukuczka Academy of Physical Education in Katowice, ul. Mikołowska 72A, 40-065 Katowice, Poland; g.zydek@awf.katowice.pl; 5Institute of Sport Sciences, Jerzy Kukuczka Academy of Physical Education in Katowice, ul. Mikołowska 72A, 40-065 Katowice, Poland; m.michalczyk@awf.katowice.pl

**Keywords:** body image, female football players, cultural differences, orthorexia nervosa, athletes, social media influence

## Abstract

Background/Objectives: Orthorexia Nervosa (ON) is an emerging behavioral pattern characterized by an obsessive focus on healthy eating. Despite its prevalence, ON lacks formal diagnostic criteria in major classification systems like the DSM-5 and the ICD-10. This study aims to investigate the impact of socio-cultural attitudes towards body image and the role of social media on the risk of ON among female football players from Poland, Turkey, and India. This study hypothesizes that socio-cultural pressures and media usage significantly influence the risk of developing ON, particularly in cultures more exposed to Western beauty ideals. Methods: The study was conducted from May to August 2024, employing the Computer-Assisted Web Interview method. A total of 142 female football players aged 16–36 from Poland, Turkey, and India participated. Data were collected using a structured questionnaire that included demographic information and health metrics, the Socio-Cultural Attitudes Towards Appearance Questionnaire, and the Duesseldorf Orthorexia Scale. Statistical analyses included an ANOVA, the Kruskal–Wallis test, the chi-square test, and Pearson’s correlation coefficient. Results: The study found that nearly half of the participants were at risk of or presented with ON, with the highest prevalence being among Indian athletes. Statistically significant relationships were observed between the risk of ON and factors such as age, dietary exclusions, social media usage, and sources of nutritional information. However, no significant correlation was found between socio-cultural attitudes and the risk of ON, suggesting that other factors may play a more critical role. Conclusions: While socio-cultural pressures and media use are contributing factors to the risk of ON, psychological factors and individual behaviors appear to be equally, if not more, significant. This study highlights the importance of targeted educational programs and psychological support for young athletes, with a focus on promoting healthy dietary practices and positive body image perceptions across varying cultural contexts. Additionally, the results suggest the need for further research into the specific psychological and behavioral mechanisms underlying ON.

## 1. Introduction

Orthorexia Nervosa (ON) involves an obsessive focus on healthy eating, in which individuals prioritize food quality over quantity, leading to restrictive diets that may cause nutritional deficiencies, malnutrition, and a reduced quality of life [1]. Despite various attempts, the diagnostic criteria for ON have not been officially endorsed, so it is not recognized as a mental disorder. As a result, ON is not classified as an eating disorder (ED) in the major systems, the ICD-10 (International Classification of Diseases) and the DSM-5 (Diagnostic and Statistical Manual of Mental Disorders) [2,3]. Both systems describe EDs, aiding clinicians in their diagnosis and treatment, but do not include ON-specific criteria [2,3]. In practice, ON is often seen as part of Other Specified Feeding or Eating Disorders (OSFEDs) or is considered a cultural variant of Anorexia Nervosa (AN) [2,3]. It is worth mentioning, however, that recent progress has been made in this field, particularly with the publication of a consensus document on the definition and diagnostic criteria for ON. A significant step forward was the article published by Donini LM et al. in 2022, which brought together researchers from different countries to establish a more unified approach to diagnosing ON [4]. However, the criteria for ON are based on expert consensus rather than empirical evidence, which poses a limitation. Additionally, the study did not include many regions of the world, limiting its global representativeness. Nonetheless, it was a breakthrough as it established preliminary, widely agreed-upon diagnostic criteria for ON, providing a foundation for future research and potentially aiding in the identification and treatment of this disorder.

From a cognitive–behavioral perspective, socio-cultural determinants are considered to be key factors influencing body image formation [5]. One of the most commonly used theories explaining the development of body image dissatisfaction is the Tripartite Influence Model. This model posits that three sources of influence—peers, parents, and media—affect body image and EDs through two mediating mechanisms: the internalization of societal appearance standards and the excessive comparison of appearance. According to this theory, social influences, such as peers, family, and media, exert pressure on individuals to conform to culturally defined beauty standards [6].

Literature reviews highlight the significant role of media in shaping physical appearance standards [7]. Research indicates that awareness, perceived pressure, and internalization significantly correlate with body image measures. However, there is confusion regarding the definitions and the operationalization of these constructs across tools. The latest tool, SATAQ-3, removed the awareness dimension, likely due to its weak association with body image or the lack of factor analysis support [8]. The awareness of cultural appearance norms does not equate to personal internalization or pressure to conform. The SATAQ-3 includes an “informational” subscale, defined by Calogero and colleagues as the “recognition that media provide information about appearance standards” [9]. However, the analysis of the questionnaire shows that the items mostly address whether media are seen as an “important source” of information about attractiveness. Positive responses may reflect both passive acknowledgment and the active seeking of media information on attractiveness standards [10].

The internalization of the thin body ideal is directly linked to body dissatisfaction and ED symptoms [11,12]. In recent years, two new discourses have emerged in Western societies: one focusing on “healthy weight” and the other on muscularity [13,14]. The healthy weight discourse, promoted by health institutions and media, emphasizes maintaining a socially acceptable body mass through a balanced diet and regular exercise, with a focus on health rather than appearance [15]. The muscularity discourse is also gaining popularity, with the internalization of muscularity increasingly influencing women [16]. Both discourses may lead to excessive body and food control, resulting in negative health outcomes [14].

This concern is particularly relevant for athletes, who often face heightened pressure to achieve both optimal physical performance and a culturally idealized body shape. The demands of maintaining peak performance, combined with the emphasis on appearance in certain sports, may uniquely interact with ON tendencies. Athletes may be especially vulnerable to developing ON, as they are encouraged to prioritize not only their physical abilities but also the quality of their diet. This dual pressure could exacerbate tendencies toward extreme dietary control and food-related anxiety, contributing to the development of ON in this population [15].

Female athletes, particularly football players, represent a unique group when studying ON and body image concerns. Unlike many sports that emphasize a thin or lean appearance, football places value on both strength and agility, making female players more likely to be influenced by both the “healthy weight” and muscularity discourses. Additionally, female athletes are often exposed to a combination of general societal pressures related to appearance and sport-specific expectations regarding their performance and body composition. These factors may make them particularly vulnerable to ED behaviors, including ON, as they strive to balance the demands of athletic performance with the societal ideals of femininity. Choosing female football players as the study population allows for an examination of how these competing pressures intersect, providing valuable insights into the ways in which sport-specific and socio-cultural influences contribute to ON development.

Until recently, body dissatisfaction was often considered a phenomenon that was primarily associated with Western countries [17]. However, the early 21st century has seen this issue spread globally, including to countries outside the Western world [18]. The rise in ED in Western Europe and the United States during the latter half of the 20th century coincided with significant social changes, such as the expansion of consumer culture and shifts in women’s social roles [19]. Several studies have highlighted that eating disorders are more prevalent in industrialized countries, such as the United States, Western Europe, and Japan, compared to preindustrial societies [20,21]. Cultures undergoing rapid social change, with strong patriarchal structures and evolving opportunities for women, are particularly vulnerable to internalizing unrealistic beauty standards, which leads to high levels of body dissatisfaction [14].

In the context of studying ON, these cultural factors interact uniquely with the pressures in athletic environments. Athletes not only face heightened expectations around performance but may also feel pressure to conform to specific body ideals, further complicating the study of ED in this population. Although EDs are observed across diverse nationalities and ethnic groups, further research is needed to understand how socio-cultural attitudes toward body appearance and the specific pressures faced by athletes shape the development of ON [22]. Directly connecting these theoretical frameworks to cultural and athletic contexts would offer greater insight into the unique challenges of studying ON in these populations.

Research shows that athletes generally have a more positive body image than non-athletes [23,24,25]. For male athletes, negative body perception is not strongly linked to EDs, but for female athletes, body dissatisfaction is a significant predictor of EDs [26,27]. Petrie and Greenleaf’s socio-cultural model suggests that socio-cultural and sport-specific pressures contribute to ED development through the internalization of appearance ideals, body dissatisfaction, and dietary restrictions [28]. Athletes face pressure to meet both general body ideals and sport-specific expectations, such as coach demands and team norms, with social media further intensifying their body image concerns [29,30].

There is a lack of comparative research on ON across different cultures, particularly within the sports context. Most existing studies focus on Western countries, leaving a gap in understanding how ON manifests in other cultures. Comparing countries with varying levels of Western cultural saturation and globalization may reveal the extent to which these influences contribute to the development of ON and how they affect socio-cultural attitudes towards body image.

This study aims to compare the impact of socio-cultural attitudes towards body image and the role of social media on the risk of ON among female football players from three different countries: Poland, Turkey, and India. The study seeks to understand the extent to which cultural differences related to body appearance contribute to the development of ON. While the hypothesis posits that Poland, a country more influenced by Western beauty standards, will show a higher risk of ON compared to Turkey and India, it is essential to acknowledge the complexity of cultural influences. Factors such as local beauty ideals, religious values, and varying levels of globalization may also play significant roles in shaping attitudes toward body image and eating behaviors.

## 2. Materials and Methods

### 2.1. The Procedure of the Survey

The study was conducted between May and August 2024, using the CAWI (Computer-Assisted Web Interview) method, which involves data collection via an online form—a recognized approach in psychological research. Google Forms was chosen for its user-friendliness, accessibility, and ability to be customized according to the needs of the study. Before the training session, the participants were provided with a QR code that directed them to the survey. To ensure correct understanding and minimize errors, researchers explained the participation process in advance. The female football players completed the survey immediately after receiving the QR code, with no time limit imposed for completing the questionnaire.

The study employed purposive sampling, meaning the sample was selected to represent specific characteristics and experiences relevant to the study’s topic. It was crucial to define precise selection criteria, such as gender, sports discipline, and nationality, to achieve the study’s objectives.

Participants were informed about the study’s purpose and anonymity and were asked to agree to the terms of data sharing. Information regarding informed and voluntary participation was provided at the beginning of the questionnaire. The study was conducted in accordance with the World Medical Association’s Declaration of Helsinki. Ethical approval was granted by the Bioethics Committee of the Silesian Medical University in Katowice (BNW/NWN/0043-3/641/35/23, date of approval: 22 September 2023), following the Law of 5 December 1996, on the Professions of Physician and Dentist (Journal of Laws 2016, item 727).

### 2.2. Study Participants

The study included 142 female football players aged 16–36 from 5 sports clubs; 2 clubs were located in Poland (n = 47), 2 in Turkey (n = 53), and 1 in India (n = 42). Based on the location of the clubs, the athletes were divided into three groups: Polish women (PL), Turkish women (TR), and Indian women (IN). The Polish players played at the 1st and 2nd levels of the competition, the Turkish players at the 3rd and 4th levels, while due to a different competition system, the Indian players participated in the Women’s State Football Championship. All the clubs involved in the study participated according to the statutes of the Polish Football Association (PZPN), the Turkish Football Federation (TFF), and the All India Football Federation (AIFF) [31,32,33]. The average age of the players was 20.74 ± 3.79.

The inclusion criteria for the study were as follows: (1) female gender, (2) consent of the club to participate in the study, (3) voluntary participation in the study, (4) aged 16 or older, (5) status as an active player at the time of the study, (6) no injury resulting in at least a 7-day break in training in the past 2 months, (7) proficiency in Polish, Turkish, or English, (8) no past or current mental illnesses including EDs diagnosed by a medical doctor. The exclusion criteria for the study were (1) a nationality other than Polish, Turkish, or Indian, and (2) an incorrectly or incompletely completed questionnaire.

### 2.3. Research Tools

This study employed a structured survey questionnaire, which included a demographic and health metrics section. This section collected detailed respondent data, such as age, height, weight, the presence of chronic diseases (including mental disorders and eating disorders), current medications, educational background, field position in football, the frequency of additional training sessions beyond club activities, sources of nutritional knowledge, dietary exclusions, and social media usage patterns. To increase the reliability of the data, demographic questions were based on standard health survey formats that are commonly used in sports research. In addition, the presence of chronic diseases and current medications taken were self-reported as open-ended questions to allow respondents to describe them accurately. Additionally, the study utilized two standardized instruments: the Socio-Cultural Attitudes Towards Appearance Questionnaire (SATAQ 3) and the Duesseldorf Orthorexia Scale (DOS).

#### 2.3.1. Body Mass Index (BMI)

The nutritional status of the participants was assessed using the BMI, which is calculated by dividing an individual’s body weight in kilograms by the square of their height in meters (BMI = weight (kg)/height (m^2^)). The resulting BMI values were interpreted based on the WHO classification. According to the WHO guidelines, a BMI of less than 18.5 is classified as underweight, a BMI between 18.50 and 24.99 is considered a normal weight, a BMI from 25.00 to 29.99 indicates being overweight, while BMI values from 30.00 to 34.99, 35.00 to 39.99, and 40.00 or higher are categorized as Grade I, Grade II, and Grade III obesity, respectively [34].

#### 2.3.2. DOS

The DOS is a screening instrument designed to evaluate ON eating behaviors. The DOS demonstrates strong internal consistency, robust construct validity, and reliable test-retest stability [35]. The ten-item version of the DOS functions as a subscale within the broader 21-item DOS, which includes three subscales: Orthorexic Eating Behavior, Additive Avoidance, and Mineral Supply. The ten-item DOS is employed as a unidimensional measure for assessing and screening ON [35].

The respondents completed the ten-item DOS using a four-point Likert scale ranging from “definitely not applicable to me” to “definitely applicable to me”, with no reverse-scored items. The maximum possible score was 40. The interpretation of the scores was as follows: a total score exceeding 30 points suggests the presence of ON, a score between 25 and 29 points indicates a risk of ON, and a score below 25 points signifies the absence of ON [35].

In this study, due to the diverse nationalities within the participant group, the research utilized the English version of the ten-item DOS (E-DOS) [35], along with Polish (PL-DOS) [36] and Turkish (Düsseldorf Ortoreksiya Ölçeği—DOÖ) [37] adaptations. The reliability of the PL-DOS and DOÖ versions was comparable to that of the E-DOS, with Cronbach’s α coefficients of 0.84 and 0.87, respectively [35,36,37]. Additionally, the Cronbach’s α coefficients calculated for the scales used in the study were as follows: 0.760 for the E-DOS scale, 0.777 for the PL-DOS scale, and 0.926 for the DOÖ scale.

#### 2.3.3. SATAQ-3

The Socio-Cultural Attitudes Toward Appearance Scale 3 (SATAQ 3) is a widely recognized tool designed to assess the influence of socio-cultural norms, especially those propagated through mass media, on individuals’ attitudes and behaviors concerning body image and physical appearance. The SATAQ 3, alongside its previous versions, functions as an effective measure for evaluating the degree to which individuals experience societal pressures and internalize these norms regarding body image. In the present study, three different versions of the SATAQ 3 were utilized: the original version created by Heinberg and Thompson [8], the Polish adaptation by Izydorczyk and Lizińczyk [38], and the Turkish adaptation by Swami and colleagues [39].

#### Original Version

Due to the reliability and validity of the questionnaire among English-speaking Indian adolescents [40], the English version of the questionnaire [8] was used in the study with Indian female athletes. India is linguistically diverse, yet English and Hindi are considered the official languages of the country (Official Languages Act of 1963). The questionnaire comprised 30 items to which respondents indicated their level of agreement using a five-point Likert scale, in which 1 denoted “strongly disagree” and 5 denoted “strongly agree”. The scale consisted of four homogeneous factors: General Internalization (9 items), Internalization—Athlete (5 items), Pressure (7 items), and Information (9 items). The Cronbach’s α coefficient for the individual subscales ranged from 0.92 to 0.96 [8].

The original version of the SATAQ scale demonstrated very good internal consistency, with McDonald’s omega reported at 0.898. Specifically, the omega values for the subscales were as follows: 0.854 for the General Internalization scale, 0.751 for the Internalization of the Athlete scale, 0.878 for the Pressure scale, and 0.846 for the Information scale.

#### Polish Version

The factor analysis results and the notably low factor loadings associated with items #3 and #9 led to their removal from the Polish adaptation of the SATAQ-3 questionnaire, which now comprises 28 items [38]. The Polish version of the SATAQ-3 identified four factors, each labeled differently from their counterparts in the original English version of the instrument. These factors include the Internalization—Pressure scale (12 items), the Internalization—Information Seeking scale (6 items), the Internalization—Athlete scale (4 items), and the Information scale (6 items). The Cronbach’s α coefficients for these subscales demonstrated satisfactory internal consistency within the Polish sample, ranging from 0.76 to 0.92 [8,38].

The Polish version of the overall SATAQ scale demonstrated excellent internal consistency, as evidenced by McDonald’s omega, which yielded a value of 0.929. Specifically, the Internalization—Pressure subscale exhibited an omega of 0.956, the Internalization—Athlete subscale had an omega of 0.826, the Internalization—Information Seeking subscale showed an omega of 0.817, and the Information subscale recorded an omega of 0.823.

#### Turkish Version

The analysis revealed that item #20 exhibited significant cross-loading across two factors, leading to its removal from the Turkish adaptation of the SATAQ-3 questionnaire, which now consists of 29 items [39]. In the Turkish version, the SATAQ-3 was structured into four distinct factors: Information (9 items), Pressure (7 items), Internalization—General (9 items), and Internalization—Athlete (4 items). The overall internal consistency of the scale was high, as indicated by McDonald’s omega, which was 0.919. The subscales demonstrated varying degrees of internal consistency, with omega values ranging from 0.739 to 0.875. Specifically, the McDonald’s omega for the Internalization—General subscale was 0.782, for the Pressure subscale was 0.875, for the Internalization—Athlete subscale was 0.739, and for the Information subscale was 0.866.

### 2.4. Statistical Analysis

The statistical analyses were conducted using Statistica v.13.3 (StatSoft, Cracow, Poland) and the R package v.4.0.0 (2020) under the GNU General Public License (The R Foundation for Statistical Computing). Quantitative data were summarized using mean values and standard deviations (X ± SD), while qualitative data were expressed as percentages.

The normality of data distribution was assessed using the Shapiro–Wilk test. Differences among female football players from different nationalities were evaluated using an Analysis of Variance (ANOVA) for comparisons involving three or more parametric groups and the Kruskal–Wallis test for comparisons involving three or more non-parametric groups. The chi-square test was employed to examine the distribution of categorical variables.

To explore the relationship between the DOS and the SATAQ-3 scores, Pearson’s correlation coefficient was calculated. This coefficient measures both the strength and direction of the linear association between the two quantitative variables.

Statistical significance was determined at a threshold of *p* < 0.05.

## 3. Results

### 3.1. Sample Characteristics

One hundred and forty-two female football players participated in the study, after taking into account the inclusion and exclusion criteria. According to nationality, the players were divided into three groups: PL—Polish women (n = 47); TR—Turkish women (n = 53); and IN—Indian women (n = 42). Nine players suffered from chronic diseases (Allergy) and six of them were on permanent medication (ex. Flixonase). The characteristics of the study group are shown in Table 1.

The principal source of nutritional knowledge among athletes was identified as the internet (n = 45; 31.7%), followed by consultations with a nutritionist (n = 37; 26.1%) and guidance from a coach (n = 29; 20.4%). The analysis revealed statistically significant variations in these sources, based on the athletes’ nationalities (*p* <0.001). Specifically, the PL group predominantly cited the internet as their primary information source (n = 24; 51.1%). In contrast, the TR cohort reported relatively equal reliance on both the internet (n = 17; 32.1%) and coaches (n = 19; 35.8%). The IN group, however, most frequently selected the nutritionist as their primary source of nutritional guidance (n = 22; 52.4%).

The majority of participants reported excluding various food groups from their diet (n = 81; 57.4%). Significant differences were observed among the groups (*p* < 0.001). Specifically, football players in the Polish (PL) group predominantly did not exclude any food groups from their diet (n = 34; 72.3%). In contrast, half of the players in the Turkish (TR) group reported excluding food groups (n = 26; 50%), whereas all participants in the Indian (IN) group excluded some food groups from their diets (n = 100; 100%).

Female athletes in the IN group were statistically significantly more likely to exclude dairy (*p* = 0.013), red meat (*p* < 0.001), and vegetables (*p* < 0.001) from their diets. Conversely, female athletes in the TR group were more likely to exclude grain products, including those that are gluten-free (*p* < 0.001). Athletes were surveyed about whether and how they modified their diets in response to physical activity. Statistically significant differences were observed, with Polish and Indian female athletes being notably more likely to affirmatively report adjusting their diets based on their physical activity (*p* < 0.001).

Additionally, the athletes were asked about their engagement in extra training sessions outside of their club commitments. Statistically significant differences were found between the groups (*p* < 0.001). The Turkish athletes most frequently reported engaging in additional training sessions 1–2 times per week (n = 21; 44.7%) or 3–4 times per week (n = 18; 38.3%). Similarly, the Polish athletes predominantly reported participating in extra sessions 1–2 times per week (n = 21; 46.7%) or 3–4 times per week (n = 17; 37.8%). The Indian athletes provided more varied responses regarding their extra training sessions. Specifically, 26.8% (n = 11) reported training occasionally or 1–2 times per week, while 19.5% (n = 8) indicated engaging in additional training 5–6 times per week or daily.

The athletes were surveyed regarding their social media activity, comparisons of their body shape with other athletes, and their body satisfaction. Statistically significant differences were observed among the groups. For detailed information, please refer to Table 2 Początek formularzaDół formularza.

### 3.2. Risk of ON

The interpretation of the DOS questionnaire results revealed that 28.2% (n = 40) of the athletes were at risk of ON, while 20.4% (n = 29) exhibited signs of ON. A statistically significant association was found between the risk or presence of ON and the athletes’ nationality (*p* < 0.001). A statistically significant relationship was also observed between nationality and the average score on the DOS scale (*p* < 0.001) (Table 3).

A statistically significant relationship was found between the mean age of the subjects and the interpretation of the DOS scale (*p* = 0.004). The female athletes without a risk of ON had an average age of 21.5 ± 0.383 years, those at risk of ON had an average age of 20.6 ± 0.732 years, and those with ON present had an average age of 19.0 ± 3.22 years. There was no significant correlation between the BMI and the interpretation of the DOS scale (*p* = 0.221). There was no significant correlation between the number of extra training units outside the club, and the DOS scale interpretation (*p* = 0.271). However, a statistically significant relationship was found between dietary adjustments and physical activity. The female football players at risk of ON (n = 35; 87.5%) or with ON present (n = 27; 93.1%) were more likely to modify their diet in response to training compared to those without an ON risk (n = 49; 67.1%) (*p* = 0.004). In addition, the female athletes who indicated their source of nutritional information to be a nutritionist were statistically significantly more likely to show the presence of ON (*p* = 0.041): as many as 40.5% of respondents who indicated this source of information had ON present, and 24.3% had an ON risk, as interpreted by the DOS scale. A correlation was found between the time spent on social media and the risk of ON (*p* = 0.035). The female athletes without an ON risk were more likely to spend more than 3 h on social media (n = 31; 42.5%) compared to those at risk of ON (n = 10; 25%) and those with ON present (n = 9; 31.0%). The female football players who used social media to acquire new information about diet and nutrition were statistically significantly more likely to exhibit ON (n = 6; 20.7%) compared to those at risk of ON (n = 3; 7.5%) or those without ON (n = 3; 4.1%) (*p* = 0.024).

Moreover, a statistically significant association was demonstrated between the risk of ON, as interpreted through the DOS scale, and satisfaction with one’s body shape (*p* = 0.003). Women who were dissatisfied with their body appearance were significantly more likely to exhibit a risk of or the presence of ON (33.3% and 33%, respectively). No statistically significant associations were found between comparing one’s body to images on social media and the risk of ON (*p* = 0.614).

There was no correlation between the DOS score and the BMI value (*p* = 0.390). Figure 1 shows the distribution of the DOS scale scores according to the subjects’ BMI and nationality.

### 3.3. Influence of Socio-Cultural Attitudes toward Appearance on ON Risks

The analysis of the SATAQ-3 results revealed no statistically significant correlations between the performance in each section and the risk of developing ON. For detailed information, please refer to Table 4.

### 3.4. The Influence of Socio-Cultural Attitudes towards Appearance on Body Comparison with Photos on Social Media and Body Satisfaction

The effects of the individual subscales of the SATAQ-3 on comparing one’s body with photos on social media and satisfaction with one’s body were analyzed across all the nationalities studied. The Polish female football players who often compared their bodies to photos exhibited statistically significantly higher scores on the Internalization—Pressure subscale (*p* < 0.001), the Internalization—Athlete subscale (*p* = 0.011), and the total scores (*p* = 0.005). The Turkish female football players who occasionally compared themselves to photos showed higher scores on the Pressure subscale (*p* = 0.035), the Internalization—General subscale (*p* = 0.027), the Internalization—Athlete subscale (*p* = 0.020), and the total scores (*p* = 0.049). The Indian female football players who frequently compared their bodies to photos on social media had statistically significantly higher scores on the Internalization—General subscale (*p* = 0.023). The Polish female football players who were dissatisfied with their bodies scored statistically significantly higher on the Internalization—Pressure subscale (*p* < 0.001), the Internalization—Athlete subscale (*p* = 0.03), and on the total scores (*p* = 0.002). No statistically significant relationship was observed among the female football players from Turkey. The Indian female football players who were satisfied with their bodies but desired some changes scored statistically significantly higher on the Internalization—General subscale (*p* = 0.014) and the total score (*p* = 0.025). Table 5 shows the detailed results.

A weak, yet statistically significant, positive correlation was observed between the scores on the DOS scale and the SATAQ3 scale, with Pearson’s correlation coefficient being calculated at r = 0.246 (*p* = 0.002). The details are shown in Figure 2.

## 4. Discussion

The results of this study align with previous research, indicating the influence of socio-cultural factors and media on the development of body image issues and eating disorders among female athletes [41,42,43]. The conducted study confirmed that nearly half of the participants were at risk of or exhibited ON, with the highest risk observed among the athletes from India, followed by Poland and Turkey. This gradient may reflect different levels of exposure to Western cultural norms and beauty ideals, which may be more internalized by athletes in cultures that are more susceptible to globalization and Western influences. This phenomenon has already been described in the literature, where it has been noted that societies undergoing rapid social and cultural changes, such as the development of a consumer economy and changes in women’s status, may be particularly prone to increased body dissatisfaction and the development of eating disorders [14,18,19].

Interestingly, the study did not show a significant correlation between socio-cultural attitudes towards body image, as measured by the SATAQ-3 questionnaire, and the risk of ON. It seems that although socio-cultural pressures exist and are an important factor influencing body perception, other factors, such as psychological predispositions and individual media usage habits, may play an equally, if not more, important role in the development of ON [44,45,46]. A weak but significant positive correlation between the SATAQ-3 and the DOS scores suggests that the internalization of social norms contributes to the risk of ON; however, it is not the only factor influencing the development of this disorder. Although it was expected that the subscales of the SATAQ-3 questionnaire, particularly those measuring the internalization of social norms and pressure, would be significantly associated with the risk of ON, the statistical analysis did not show such a direct correlation. This implies that other factors, such as individual psychological predispositions (e.g., perfectionism and anxiety) or more subtle, contextual behavioral factors (e.g., the manner and intensity of social media use), may play a decisive role in the development of this disorder. This highlights the complexity of the ON issue and the need for further research that includes a broader range of variables, including psychological and behavioral contexts. While socio-cultural pressure and societal expectations regarding body image are important, they cannot be regarded as the sole factors leading to ON [6,10,11].

The significant role of social media in shaping body image and eating behaviors is also confirmed by other studies. For instance, Tiggemann and Slater [47] and Fardouly and Vartanian [48] demonstrated that the time spent on social media is positively correlated with body dissatisfaction and the internalization of the thin ideal. The study found that athletes who frequently compared their bodies to photos on social media or were dissatisfied with their appearance exhibited higher internalization of socio-cultural norms. This is consistent with the Tripartite Influence Model, which posits that media, combined with peers and family, play a key role in shaping body image [6]. However, the lack of a direct correlation between comparisons on social media and the risk of ON suggests that social media may exacerbate body dissatisfaction but does not necessarily lead directly to ON unless other psychological or behavioral factors are also present [49].

Another significant finding is the difference in the sources of nutritional knowledge and dietary practices among athletes from different nationalities. The reliance on the internet as the main source of nutritional information among the athletes from Poland and Turkey contrasts with the preference for consulting dietitians that was identified among the athletes from India. This difference may result from cultural differences in access to and trust in various information sources, which may also influence the development of ON. Misinformation or an excessive focus on “clean eating” being promoted by internet sources could contribute to restrictive eating behaviors, as confirmed by the studies of Turner and Lefevre [50] and Koven and Abry [1], which highlight the role of the internet and social media in promoting orthorexic eating patterns. This study, however, showed that theIndian athletes most frequently identified dietetic consultations as their source of nutritional information, which in turn was associated with an increased risk of or the presence of ON. Athletes who regularly consult with dietitians may be more aware and interested in healthy eating, which can lead to an excessive focus on “clean eating” and perfectionism in their dietary choices. In the context of ON, this excessive focus and striving for perfection may turn into an obsession, in which every dietary choice is analyzed for its “purity” and health impact. Moreover, in countries like India, where dietary norms may be strongly rooted in tradition, athletes may feel pressured to fully adhere to dietitian recommendations, which, combined with the influence of media promoting ideal “clean” diets, may lead to the restrictive eating behaviors that are typical of ON [51].

Additionally, the study highlighted the significance of age as a risk factor, with younger athletes being more susceptible to ON. These findings are consistent with the literature, which indicates that adolescence and early adulthood are critical periods for the development of ED, resulting from increased sensitivity to social pressure and body image issues [27,29,52]. Brasil et al. demonstrated that younger individuals, particularly women, are more vulnerable to the influence of media and society in shaping body image, which can lead to the development of eating disorders [53].

This study partially confirmed the hypothesis that socio-cultural influences and social media use play a role in the risk of developing ON among female football players from Poland, Turkey, and India; however, the results were not unequivocal. The hypothesis that players from cultures more exposed to Western beauty ideals, such as Poland, would exhibit a higher risk of ON was not supported, as it was the players from India, rather than Poland, who showed the highest risk of ON. This suggests that, in addition to socio-cultural influences and media, other factors such as traditional dietary practices and the growing impact of Western media in countries like India may play a more significant role in the development of ON. Ultimately, while socio-cultural influences are important, this study indicates that exposure to Western beauty ideals alone is insufficient to explain the risk of ON, highlighting the complexity of this disorder and the need for further research that considers a broader range of factors.

Given the study’s findings, future research should focus on understanding the relationship between psychological predispositions, media consumption patterns, and socio-cultural pressure in the context of ON development. Additionally, there is a need to examine the role of traditional and digital media in shaping body image among athletes from different cultures, particularly in non-Western contexts where cultural norms are rapidly changing.

### Strengths and Limitations

This study included female football players from three different countries (Poland, Turkey, and India), which allowed for the analysis of the influence of various socio-cultural norms on the risk of developing ON. This broader context enhanced the applicability of the study’s findings across different cultures. The use of validated tools, such as the SATAQ-3 and the DOS, increased the reliability and accuracy of the collected data. The focus on professional female football players allowed for a more detailed analysis of the risk factors specific to this group concerning ON. Additionally, this study not only assessed the risk of ON but also investigated the dominant sources of nutritional information (internet, dietitians, trainers) among the athletes, providing valuable insights into potential risk factors.

However, the study also has some limitations. Despite the geographical diversity, the study included a relatively small number of participants (142 athletes), which may limit the generalizability of the findings to broader populations. The use of purposive sampling may have introduced a bias, as the sample was not randomly selected, potentially limiting the generalizability of the findings to the broader population of female football players. Although the CAWI (Computer-Assisted Web Interview) method is convenient and widely used, it may have certain limitations; however, the authors made every effort to minimize potential errors, such as providing the opportunity to obtain direct explanations from the researchers. It is also important to note that there are many potential factors influencing the risk of ON, and the study does not account for all the possible variables. Additionally, the methods section did not mention controlling for potential confounding variables, such as socio-economic status, training intensity, and dietary habits, which could influence the outcomes and impact the study’s internal validity. Nevertheless, this study is one of the first to assess the risk of ON across multiple nationalities, additionally considering socio-cultural factors.

## 5. Conclusions

This study revealed that the primary source of nutritional information among the athletes was the internet, particularly among the football players from Poland and Turkey. Conversely, the athletes from India more frequently sought advice from dietitians. Significant differences were noted in the exclusion of food groups between the cohorts. The Indian athletes were more likely to exclude dairy, red meat, and vegetables from their diet, which may be associated with cultural dietary practices and beliefs.

The amount of time spent on social media and the types of platforms used varied between the groups. The Turkish football players spent the most time on social media, followed by the Polish athletes, with the Indian athletes spending the least amount of time on social media. The Indian athletes were more inclined to spend time on YouTube, whereas the Polish and the Turkish players were more active on Instagram.

Nearly 50% of the surveyed athletes exhibited a risk of or the presence of ON. The Turkish athletes demonstrated the lowest risk among the nationalities studied, followed by the Polish players, while the Indian football players were the most at risk. Age was a significant factor—the younger athletes were more vulnerable to ON. Similarly, the exclusion of various food groups from their diet, as well as the use of social media to seek information on nutrition and diets, constituted risk factors for the development of ON. In addition, the female athletes who identified their source of nutritional information as a nutritionist were more likely to report the presence of or the risk of ON. No significant correlation was found between the BMI and the ON risk, suggesting that the ON risk is more closely associated with psychological factors than with physical attributes such as body weight. The study found that the athletes who were dissatisfied with their appearance were more likely to be at a higher risk for, or to exhibit, ON, as interpreted by the DOS scale.

Furthermore, this study indicated that socio-cultural attitudes towards body image did not have a significant impact on the risk of developing Orthorexia Nervosa among the female football players of all the nationalities studied. A weak, yet statistically significant, positive correlation was observed between the SATAQ-3 and the DOS scores. However, the study did not demonstrate a significant direct impact of the SATAQ-3 subscales on the ON risk, indicating that although socio-cultural attitudes play a role, other factors also contribute to the development of ON.

This study also suggested that athletes who frequently compared themselves to others on social media or were dissatisfied with their bodies exhibited a higher internalization of socio-cultural norms.

A practical conclusion from this study is the need to implement educational programs and psychological support within the sports environment that promote a healthy approach to nutrition and body image, particularly among younger athletes. These programs should take cultural diversity into account and counteract the influence of social media on body perception. Recognizing the varying influences of socio-cultural pressures across different countries and the unique challenges faced by athletes from diverse backgrounds, it would be useful to design culturally sensitive interventions. For example, programs in regions where traditional dietary practices are prevalent could emphasize the balance between cultural norms and healthy eating practices, while those in more Westernized settings could focus on combating the pressures of media-driven body ideals. Such targeted strategies would likely be more effective in addressing the specific needs of athletes within their respective cultural frameworks. Support from nutrition specialists and sports psychologists will be crucial for enhancing the physical and mental health, as well as the overall performance, of female football players.

## Figures and Tables

**Figure 1 nutrients-16-03199-f001:**
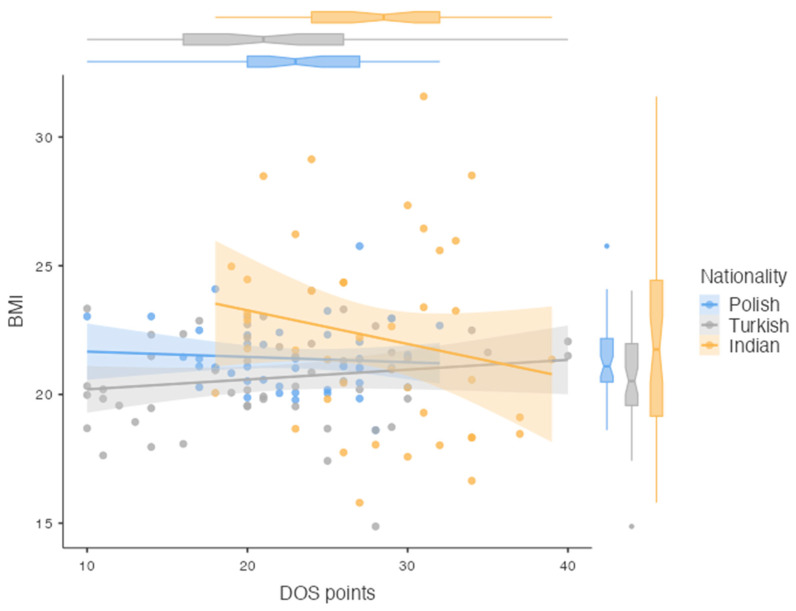
Scatter plots showing the sum of the scores obtained in the DOS with the BMI, by the nationality of players (n = 142).

**Figure 2 nutrients-16-03199-f002:**
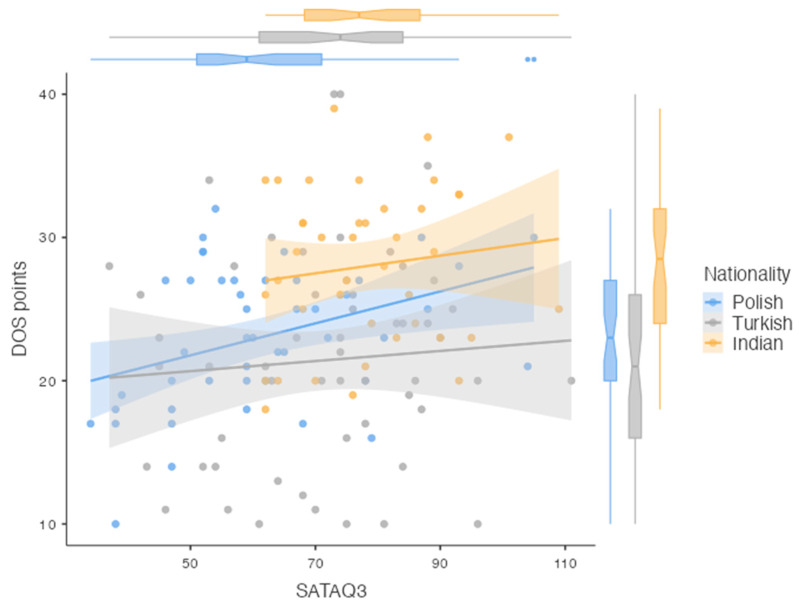
Statistical analysis of the association of the SATAQ-3 and the DOS test scores (n = 142).

**Table 1 nutrients-16-03199-t001:** Characteristics of the study group (n = 137).

	Age [Years] (X ± SD)	Height [cm] (X ± SD)	Body Mass [kg] (X ± SD)	BMI [kg/m^2^] (X ± SD)
Total (n = 142)	20.7 ± 3.79	163.0 ± 7.90	56.8 ± 7.93	21.4 ± 2.50
PL (n = 47)	22.0 ± 4.10	168.6 ± 5.49	60.1 ± 5.49	21.4 ± 1.29
TR (n = 53)	21.5 ± 2.70	164.8 ± 6.10	56.9 ± 6.05	20.6 ± 1.80
IN (n = 42)	18.4 ± 3.63	155.5 ± 6.97	54.0 ± 10.69	22.2 ± 3.75
*p*-value	<0.001 *	<0.001 *	<0.001 *	0.012 *

X—average; SD—standard deviation; PL—Polish athletes; TR—Turkish athletes; IN—Indian athletes; *—*p* < 0.05.

**Table 2 nutrients-16-03199-t002:** Activity of female footballers on social media by nationality (n = 142).

Nationality	PL (n = 47)	TR (n = 53)	IN (n = 42)	Total (n = 142)	*p*-Value
Time of use of social media during the day. n (%)					
Up to 1 h	2 (4.3)	4 (7.5)	10 (23.8)	16 (11.3)	0.004 *
1–2 h	17 (36.2)	11 (20.8)	15 (35.7)	43 (30.3)	
2–3 h	17 (36.2)	16 (30.2)	5 (11.9)	38 (26.8)	
above 3 h	11 (23.4)	22 (41.5)	12 (28.6)	45 (31.7)	
The most common type of application. n (%)					
Facebook	5 (10.6)	1 (1.9)	8 (19.0)	14 (9.9)	<0.001 *
Instagram	31 (66.6)	44 (83.0)	5 (11.9)	80 (56.3)	
Snapchat	0	2 (3.8)	0	2 (1.4)	
TikTok	7 (14.9)	0	0	7 (4.9)	
WhatsApp	0	1 (1.9)	10 (23.8)	11 (7.7)	
X (Twitter)	1 (2.1)	2 (3.8)	0	3 (2.1)	
YouTube	3 (6.4)	3 (5.7)	19 (45.2)	25 (17.6)	
Purpose of using social media. n (%)					
To relax.	23 (48.9)	18 (34.0)	10 (23.8)	51 (35.9)	0.045 *
I am looking for information on sports.	3 (6.4)	10 (18.9)	3 (7.1)	16 (11.3)	0.086
I look for information on diet/nutrition.	2 (4.3)	2 (3.8)	8 (19.0)	12 (8.5)	0.013 *
I look for the news of the day.	3 (6.4)	11 (20.8)	19 (45.2)	33 (23.2)	<0.001 *
I check what is going on with friends.	16 (34.0)	13 (24.5)	2 (4.8)	31 (21.8)	0.003 *
Comparing body image to social media photos. n (%)					
No, never.	23 (48.9)	22 (41.5)	29 (69.0)	74 (52.1)	0.004 *
Yes, sometimes.	20 (42.6)	30 (56.6)	8 (19.0)	58 (40.8)	
Yes, often.	4 (8.5)	1 (1.9)	5 (11.9)	10 (7.0)	
Satisfied with the appearance of body. n (%)					
I am not satisfied, there are many things I would like to change.	6 (12.8)	1 (1.9)	8 (19.0)	15 (10.6)	<0.001 *
Yes, but I would change a few things about my appearance.	29 (61.7)	29 (54.7)	7 (16.7)	65 (45.8)	
Yes, I would not change a thing.	12 (25.5)	23 (43.4)	27 (64.3)	62 (43.7)	

PL—Polish athletes; TR—Turkish athletes; IN—Indian athletes; *—*p* < 0.05.

**Table 3 nutrients-16-03199-t003:** Summary of ON risk estimation (DOS) (n = 142).

DOS	PL (n = 47) n (%)	TR (n = 53) n (%)	IN (n = 42) n (%)	Total (n = 142) n (%)	*p*-Value
No risk ≤ 25	25 (53.2)	36 (67.9)	12 (28.6)	73 (51.4)	<0.001 *
Risk 25–29	18 (38.3)	11 (20.8)	11 (26.2)	40 (28.2)	
Presence ≥ 30	4 (8.5)	6 (11.3)	19 (45.2)	29 (20.4)	
	PL (n = 47)	TR (n = 53)	IN (n = 42)	Total (n = 142)	
Points (X ± SD)	23.1 ± 4.9	21.4 ± 7.5	28.0 ± 5.5	23.9 ± 6.7	<0.001 *

X—average; SD—standard deviation; PL—Polish athletes; TR—Turkish athletes; IN—Indian athletes; *—*p* < 0.05.

**Table 4 nutrients-16-03199-t004:** Comparison of the SATAQ-3 scores between Polish, Turkish, and Indian female football players, based on the risk of developing ON (DOS).

**PL (n = 47) X ± SD**					
**Scale (Point)**	**Total (n = 47)**	**No Risk (n = 25)**	**Risk (n = 18)**	**Presence (n = 4)**	***p*-Value**
I-P (12–60)	21.3 ± 9.97	19.5 ± 9.07	22.2 ± 9.50	28.3 ± 16.09	0.389
I-IS (6–30)	18.1 ± 6.02	17.0 ± 5.34	18.6 ± 6.69	22.5 ± 4.80	0.122
IA (4–20)	10.9 ± 4.15	9.9 ± 3.93	12.28 ± 4.28	11.0 ± 4.16	0.179
I (6–30)	11.6 ± 4.32	11.4 ± 4.40	11.8 ± 3.72	11.8 ± 7.23	0.840
Total (28–140)	62.0 ± 16.6	57.8 ± 16.60	64.9 ± 13.0	70.5 ± 25.9	0.213
**TR (n = 53) X ± SD**					
**Scale (Point)**	**Total (n = 53)**	**No Risk (n = 36)**	**Risk (n = 11)**	**Presence (n = 6)**	***p*-Value**
P (7–35)	15.1 ± 5.35	15.0 ± 5.67	14.8 ± 5.12	16.0 ± 4.43	0.865
I-G (9–45)	20.2 ± 6.51	20.1 ± 6.26	19.5 ± 7.33	22.3 ± 7.23	0.593
I-A (4–20)	11.5 ± 2.92	11.72 ± 3.27	11.82 ± 1.83	9.33 ± 0.82	0.114
I (9–45)	24.4 ± 6.22	24.7 ± 6.23	24.2 ± 7.95	23.2 ± 1.47	0.415
Total (29–145)	71.2 ± 16.1	71.5 ± 16.4	70.3 ± 18.2	70.8 ± 11.8	0.962
**IN (n = 42) X ± SD**					
**Scale (Point)**	**Total (n = 42)**	**No Risk (n = 12)**	**Risk (n = 11)**	**Presence (n = 19)**	***p*-Value**
P (7–35)	15.6 ± 4.47	16.8 ± 3.91	14.1 ± 5.07	15.7 ± 4.42	0.170
I-G (9–45)	24.0 ± 3.37	25.1 ± 3.15	23.8 ± 3.34	23.5 ± 3.53	0.278
I-A (5–25)	13.0 ± 3.69	12.4 ± 3.96	12.8 ± 3.31	13.6 ± 3.81	0.165
I (9–45)	25.5 ± 3.54	23.9 ± 2.61	26.4 ± 5.14	26.1 ± 2.68	0.165
Total (30–150)	78.2 ± 11.6	78.2 ± 11.8	77.1 ± 13.6	78.8 ± 10.8	0.756

X—average; SD—standard deviation; PL—Polish athletes; TR—Turkish athletes; IN—Indian athletes; I-P: Internalization—Pressure; I-IS: Internalization—Information Seeking; I-A: Internalization—Athlete; I: Information; P: Pressure; I-G: Internalization—General.

**Table 5 nutrients-16-03199-t005:** The comparison of the SATAQ-3 scores between the Polish, Turkish, and Indian female football players, based on comparing their bodies to social media photos and satisfaction with their bodies.

**Comparing Body Image to Social Media Photos**				
PL (n = 47) X ± SD				
Scale	No, never (n = 23)	Yes, occasionally (n = 20)	Yes, often (n = 4)	*p*-value
I-P	16.43 ± 6.56	25.06 ± 10.95	30.50 ± 7.85	<0.001 *
I-IS	17.35 ± 6.60	18.95 ± 5.34	18.75 ± 6.85	0.527
I-A	9.13 ± 4.00	12.60 ± 3.70	12.75 ± 3.40	0.011 *
I	11.17 ± 4.31	11.95 ± 4.71	12.25 ± 2.50	0.724
Total	54.09 ± 12.97	68.55 ± 16.98	74.25 ± 15.50	0.005 *
TR (n = 53) X ± SD				
Scale	No, never (n = 22)	Yes, occasionally (n = 30)	Yes, often (n = 1)	*p*-value
P	13.27 ± 4.43	16.60 ± 5.57	9.00 ± 0	0.035 *
I-G	17.95 ± 6.28	22.10 ± 6.13	12.00 ± 0	0.027 *
I-A	10.36 ± 2.89	12.40 ± 2.65	8.00 ± 0	0.020 *
I	24.64 ± 7.14	24.63 ± 5.35	14.00 ± 0	0.289
Total	66.23 ± 14.98	75.73 ± 15.30	43.00 ± 0	0.049 *
IN (n = 42) X ± SD				
Scale	No, never (n = 29)	Yes, occasionally (n = 8)	Yes, often (n = 5)	*p*-value
P	15.6 ± 4.68	15.3 ± 4.20	16.0 ± 4.47	0.968
I-G	23.7 ± 2.91	23.3 ± 4.65	27.4 ± 1.67	0.023 *
I-A	12.7 ± 3.28	12.9 ± 4.49	15.2 ± 4.60	0.463
I	25.8 ± 3.63	25.6 ± 3.38	24.0 ± 3.54	0.490
Total	77.8 ± 11.17	77.0 ± 14.26	82.6 ± 10.88	0.522
**Satisfaction with the Appearance of Your Body Image**				
PL (n = 47) X ± SD				
Scale	Not satisfied (n = 6)	Yes, but wants to change a few things (n = 29)	Yes, nothing to change (n = 12)	*p*-value
I-P	30.50 ± 10.27	22.62 ± 9.79	13.50 ± 2.61	<0.001 *
I-IS	18.83 ± 4.31	18.69 ± 6.02	16.5 ± 6.86	0.389
I-A	12.83 ± 3.25	11.59 ± 4.15	8.33 ± 3.58	0.030 *
I	13.33 ± 4.63	11.79 ± 4.34	10.25 ± 4.07	0.377
Total	75.50 ± 19.26	64.69 ± 15.02	48.58 ± 9.87	0.002 *
TR (n = 53) X ± SD				
Scale	Not satisfied (n = 1)	Yes, but wants to change a few things (n = 29)	yes, nothing to change (n = 23)	*p*-value
P	16.0 ± 0	16.3 ± 5.15	13.5 ± 5.43	0.089
I-G	24.0 ± 0	21.5 ± 6.10	18.3 ± 6.81	0.218
I-A	10.0 ± 0	12.1 ± 2.77	10.7 ± 3.03	0.114
I	23.0 ± 0	24.4 ± 6.26	24.6 ± 6.44	0.875
Total	73.0 ± 0	74.3 ± 16.32	67.2 ± 15.59	0.253
IN (n = 42) X ± SD				
Scale	Not satisfied (n = 8)	Yes, but wants to change a few things (n = 7)	yes, nothing to change (n = 27)	*p*-value
P	17.1 ± 4.73	17.1 ± 3.08	14.7 ± 4.60	0.149
I-G	24.1 ± 4.26	27.0 ± 1.91	23.2 ± 3.02	0.014 *
I-A	14.0 ± 4.07	15.6 ± 3.41	12.1 ± 3.36	0.067
I	25.6 ± 3.29	26.9 ± 3.59	25.1 ± 3.63	0.518
Total	80.9 ± 13.76	86.6 ± 6.27	75.2 ± 11.02	0.025*

X—average; SD—standard deviation; PL—Polish athletes; TR—Turkish athletes; IN—Indian athletes; *—*p* < 0.05; I-P: Internalization—Pressure; I-IS: Internalization—Information Seeking; I-A: Internalization—Athlete; I: Information; P: Pressure; I-G: Internalization—General.

## Data Availability

The raw data supporting the conclusions of this article will be made available by the authors on request.

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
