# Peer review of "The Impact of Social Media and Socio-Cultural Attitudes toward Body Image on the Risk of Orthorexia among Female Football Players of Different Nationalities"

_nutrients, 2024, doi:10.3390/nu16183199_

Round 1

Reviewer 1 Report

Comments and Suggestions for Authors

Please check the attachment

Comments on the Quality of English Language

I suggest editing the English throughout the MS. This is true for both the grammar and for making the text more concise.

Author Response

Thank you so much for taking the time to evaluate our work. We have tried to incorporate all your valuable suggestions. If we could improve our work in any way, please let us know.

Comment 1

The abstract effectively identifies Orthorexia Nervosa (ON) as a behavioural pattern lacking formal diagnostic criteria. It explores its relationship with socio-cultural attitudes and social media use among female football players from Poland, Turkey, and India. While the study's cross-cultural focus and use of validated tools are strengths, the relatively small sample size (142 participants) may limit the generalizability of the findings. The results reveal a high prevalence of ON, particularly among Indian athletes, with significant correlations between ON risk and factors like age, dietary exclusions, and social media usage. However, the lack of a significant correlation with socio-cultural attitudes, despite being a primary hypothesis, suggests that other psychological or individual factors might play a more critical role. This point could be better addressed. The conclusion underscores the need for targeted educational and psychological support across different cultural contexts but could benefit from recommendations for future research and a discussion of the study's limitations.

Thank you very much for your suggestions, the summary has been corrected.

Comment 2

Introduction Pros:

  1. The introduction provides a thorough background on ON, highlighting its key characteristics, the associated health risks, and the lack of formal diagnostic criteria. This sets a strong foundation for the study and emphasizes the importance of further research in this area.
  2. The text effectively integrates cognitive-behavioural theories and socio-cultural models, such as the Tripartite Influence Model and the socio-cultural model proposed by Petrie and Greenleaf. This theoretical grounding helps explain how socio-cultural factors might influence body image and ON, making the research focus more robust.
  3. The introduction identifies a clear research gap in comparative studies on ON across different cultures, particularly within the sports context. By focusing on female football players from Poland, Turkey, and India, the study aims to explore how varying degrees of Western cultural influence might impact the development of ON, which is both novel and relevant.
  4. The hypothesis is clearly stated, positing that socio-cultural attitudes and social media usage will significantly influence the risk of ON, particularly in cultures more exposed to Western beauty ideals. This clarity helps guide the reader through the purpose of the study.

Cons:

  1. The introduction is quite lengthy and dense, which might overwhelm the reader. While the content is rich and informative, it could benefit from more concise language and a streamlined structure to improve readability and focus. Some sections, particularly those discussing tools like the SATAQ-3, could be summarized without losing essential details.
  2. While the theoretical background is strong, it takes up a significant portion of the introduction. This could be balanced by providing more context on the specific challenges and nuances of studying ON in the chosen cultural and athletic contexts. A more direct connection between the theoretical models and the population under study might make the introduction more cohesive.
  3. The introduction touches on body dissatisfaction and eating disorders in athletes but could expand more on why ON specifically might be a concern in this population. A deeper exploration of how the pressures of athletic performance and appearance might uniquely interact with ON would strengthen the rationale for the study.
  4. The hypothesis assumes that Poland, being more saturated with Western beauty standards, will show a higher risk of ON compared to Turkey and India. This bold claim could benefit from more nuance, acknowledging that cultural influences are complex

and multifaceted. The introduction could also discuss potential counterarguments or

alternative explanations for cultural differences in ON prevalence.

  1. The choice of female football players as the study population is mentioned but not thoroughly justified. A brief discussion on why this specific group was chosen and how they might be particularly affected by the factors under study would add depth to the

introduction.

Thank you very much for your insightful analysis, we have taken into account all the suggestions by improving the introduction of the survey. We hope it is sufficient for you.

Comment 3

Materials and Methods Pros:

  1. The survey procedure is well-defined, including the method (CAWI), tool (Google Forms), and steps to ensure accurate and error-free responses. This clarity helps in understanding the study design and replicating the study if needed.
  2. The study adheres to ethical guidelines, with approval from the Bioethics Committee and compliance with the Declaration of Helsinki. This ensures that the research is conducted respecting participant rights and well-being.
  3. The study employs well-established instruments like the DOS and SATAQ-3, validated in multiple languages and contexts. This strengthens the reliability and validity of the data collected.
  4. The inclusion and exclusion criteria are specific, helping to ensure that the sample is relevant to the research objectives. This minimizes potential confounding variables.
  5. The study includes participants from three different countries (Poland, Turkey, and India), which enhances the generalizability of the findings across different cultural contexts.
  6. Using appropriate statistical tests (ANOVA, Kruskal-Wallis, chi-square, Pearson's correlation) to analyze the data demonstrates a sound understanding of the relationships between variables.

Cons:

  1. Using purposive sampling may introduce bias, as the sample is not randomly selected. This could limit the generalizability of the results, as the sample may not fully represent the broader population of female football players.
  2. The methods do not mention controlling for potential confounders (e.g., socioeconomic status, training intensity, dietary habits) that could influence the outcomes, affecting the study's internal validity.
  3. The description of the demographic and health metrics section is somewhat general. More detail on the specific questions asked and how they were developed or validated would provide a clearer understanding of the data's reliability.
  4. Although the study used translations of the DOS and SATAQ-3, there is no mention of how language differences were addressed regarding cultural adaptation or potential misunderstandings, especially since English is not the first language for most participants.
  5. The sample size of 142 participants, divided among three groups, may be too small to detect significant differences in some comparisons, particularly when using ANOVA, which requires larger sample sizes to achieve sufficient power.
  6. Although the study includes participants from three countries, selecting just one club from India might not capture the diversity within that country, potentially limiting the applicability of the findings to other regions within India.

The description of the methodology was corrected as suggested, and the description of the study's weaknesses was modified to include the concerns identified by the Reviewer. Thank you very much for this guidance. The DOS and SATAQ3 questionnaires used in the study were validated in the native languages of the respondents, ensuring that the content was clear and understandable to the participants. The validation of the questionnaires is cited in the methods section, and the questionnaires are available after translation (the 'forward-backward' translation procedure was used). To increase the reliability of the ANOVA analysis, very homogeneous study groups were used, and precise inclusion and exclusion criteria were defined to eliminate some confounding factors.One club from India was recruited for the study; however, it has two teams. In the study, an effort was made to ensure comparability of results, also by having equally sized study groups.

Comment 4

Results Pros:

  1. The study provides a detailed breakdown of the sample characteristics, including nationality, chronic conditions, and sources of nutritional knowledge. This allows for a

clear understanding of the demographic composition and the context in which the

findings were made.

  1. The results include a thorough analysis of statistical significance for various factors,

such as nationality, dietary habits, and social media usage. This strengthens the validity

of the findings and allows for a more nuanced interpretation of the data.

  1. The study effectively identifies significant relationships, such as the correlation between the risk of orthorexia nervosa (ON) and factors like age, nutritional information sources, social media use, and body satisfaction. These findings contribute

valuable insights into the factors that may influence ON risk among female athletes.

  1. The results offer a comparative analysis of different nationalities, which adds depth to the understanding of how cultural differences might impact nutritional behaviours and

the risk of ON.

  1. The study examines sociocultural attitudes towards appearance and their impact on

body image and ON risk. This is a strength as it provides a broader context for

understanding the psychological factors at play.

Cons:

  1. The division into three nationality-based subgroups (with sample sizes of 47, 53, and 42) may reduce statistical power, especially for detecting subtle differences or interactions between variables.
  2. While the study identifies relationships between various factors and ON risk, it does not fully account for potential confounding variables, such as socio-economic status, level of education, or access to health services, which might influence nutritional behaviours and ON risk.
  3. Some areas, such as the influence of chronic diseases or permanent medication use on nutritional behaviours or ON risk, are mentioned but not explored in depth. This limits the understanding of how these factors might interact with the other variables studied.
  4. The study’s focus on the SATAQ-3 scale for sociocultural attitudes may not capture all relevant cultural influences on body image and ON risk, potentially overlooking other important sociocultural factors..

Thank you very much for your insightful review. The study provides unique data from three different countries (Poland, Turkey, India), which allows us to compare cultural and social influences on ON risk. Even with smaller samples, the results suggest important trends that can provide a starting point for further, more extensive research. In addition, to increase the reliability of the ANOVA analysis, very homogeneous study groups were used, and inclusion and exclusion criteria were carefully defined to eliminate some confounding factors. Participants were asked questions regarding the use of medications and the occurrence of chronic diseases. However, the very small number of individuals reporting these variables made proper analysis of the results impossible. Female soccer players, as athletes, are generally healthy, and for this reason, such factors were not analyzed in depth. Despite the limitations of the SATAQ-3 scale, the study provides important insights into the impact of socio-cultural norms and social media on ED (eating disorders). The SATAQ-3 scale has been validated as a reliable tool for assessing social influences, and its use in this study allows for comparable results with other studies in this field. Additionally, the findings indicate that other factors, such as social media use, may have a greater impact on the risk of ED than social norms alone. This study represents an important step in examining the complex interactions between social, cultural, and psychological risk factors for ED.

Comment 5

Discussion Pros:

  1. The study's inclusion of female football players from three countries (Poland, Turkey, and India) is a significant strength. This broader context enhances the understanding of how various socio-cultural norms influence the risk of developing Orthorexia Nervosa (ON), providing applicable insights across different cultures.
  2. Using the SATAQ-3 and DOS adds reliability and accuracy to the data collected, ensuring that the findings are based on validated measures.
  3. By focusing on professional female football players, the study addresses a specific population at risk for ON. This allows for a more detailed analysis of risk factors relevant to this group.
  4. Examining the sources of nutritional information (internet, dietitians, trainers) among athletes is valuable, offering insights into potential risk factors for ON, particularly the role of misinformation from non-expert sources.
  5. The study acknowledges the complexity of ON by considering sociocultural factors alongside psychological predispositions and media consumption patterns, indicating a holistic approach to understanding the disorder.
  6. Highlighting the significance of age, with younger athletes being more susceptible to ON, which aligns with existing literature on the development of eating disorders during adolescence and early adulthood.

Cons:

  1. The study's findings on the role of sociocultural attitudes towards body image in the development of ON are not unequivocal. The lack of a significant direct impact of SATAQ-3 subscales on ON risk suggests that other factors may play a more prominent role, making it difficult to draw definitive conclusions about the influence of sociocultural pressures. This should be further outlined in the discussion.
  2. The study acknowledges the complexity of ON but may still oversimplify the interplay of risk factors by not fully exploring the potential interactions between psychological, behavioural, and socio-cultural influences.
  3. While the study concludes with recommendations for educational programs and psychological support, it could have provided more specific strategies tailored to different cultural contexts rather than general suggestions...

Thank you very much, the discussion and the results of the survey have been corrected according to your recommendations..

Comment 6

General comments:

  1. I suggest editing the English throughout the MS. This is true for both the grammar and for making the text more concise.
  2. I find tables 4 and 5 overloaded. I suggest making them more friendly to the reader..

Table 4 and 5 have been revised to be more readable. In addition, making linguistic corrections throughout the manuscript.

Thank you for your help. Your guidance is invaluable.

Kind regards,

Authors.

Reviewer 2 Report

Comments and Suggestions for Authors

Title: The Impact of Sociocultural Attitudes Toward Body Image and  Social Media on the Risk of Orthorexia Among Female Football 3 Players of Different Nationalities

Thank you for the opportunity to review this manuscript as it falls within the area of my research.

I find this particular topic very interesting as we continue to research for different factors or predisposing factors of orthorexic behaviors that will help us to identify more clearly the intricacies of this emerging disorder.

Minor Revisions:

-In the first paragraph of the introduction, the authors should make reference to the fact that although there is no clear diagnosis about ON, there is a recently published article on a consensus among researchers from all the countries that have dealt with the diagnosis of orthorexia and that I believe should be mentioned in the introduction.

“Donini LM et al. A consensus document on definition and diagnostic criteria for orthorexia nervosa. Eat Weight Disord. 2022 Dec;27(8):3695-3711. doi: 10.1007/s40519-022-01512-5. Epub 2022 Nov 27. Erratum in: Eat Weight Disord. 2023 Sep 16;28(1):76. doi: 10.1007/s40519-023-01599-4. PMID: 36436144; PMCID: PMC9803763.”

- In line 117, the following statement needs references to assist the reader  “It is well documented that the prevalence of eating disorders has been higher in industrialized or  postindustrial countries, such as the United States, Western Europe, and Japan, compared  to preindustrial societies”

-Line 298, there must be an error in this sentence-, “According to innationality….”

-Table 3 and 4 are very confusing, as all the data are mixed up in the presentation form, it is not easy to see the data and finally it is not well understood.

-Line 538, says “youger athletes” and should be “younger..”

Comments on the Quality of English Language

Despite not being a native English speaker, I have noticed a few language-related issues that could benefit from further revision to ensure clarity and accuracy.

Author Response

Thank you so much for taking the time to evaluate our work. We have tried to incorporate all your valuable suggestions. If we could improve our work in any way, please let us know.

Comment 1

-In the first paragraph of the introduction, the authors should make reference to the fact that although there is no clear diagnosis about ON, there is a recently published article on a consensus among researchers from all the countries that have dealt with the diagnosis of orthorexia and that I believe should be mentioned in the introduction.

“Donini LM et al. A consensus document on definition and diagnostic criteria for orthorexia nervosa. Eat Weight Disord. 2022 Dec;27(8):3695-3711. doi: 10.1007/s40519-022-01512-5. Epub 2022 Nov 27. Erratum in: Eat Weight Disord. 2023 Sep 16;28(1):76. doi: 10.1007/s40519-023-01599-4. PMID: 36436144; PMCID: PMC9803763.”

Thank you very much for your suggestion, which will certainly affect the quality of our research. The introduction has been supplemented with the indicated information..

Comment 2

- In line 117, the following statement needs references to assist the reader  “It is well documented that the prevalence of eating disorders has been higher in industrialized or  postindustrial countries, such as the United States, Western Europe, and Japan, compared  to preindustrial societies”

Corrected as suggested..

Comment 3

-Line 298, there must be an error in this sentence-, “According to innationality….”

Sorry for the typing error, corrected..

Comment 4

-Table 3 and 4 are very confusing, as all the data are mixed up in the presentation form, it is not easy to see the data and finally it is not well understood.

..

Corrected the table to be more readable.

Comment 5

-Line 538, says “youger athletes” and should be “younger..”

A typing error has been corrected.

Comment 6

Despite not being a native English speaker, I have noticed a few language-related issues that could benefit from further revision to ensure clarity and accuracy.

Linguistic corrections have been made throughout the manuscript.

Thank you for your help. Your guidance is invaluable.

Kind regards,

Authors.

Round 2

Reviewer 1 Report

Comments and Suggestions for Authors

The authors have responded to the raised concerns. Therefore, the MS can be accepted.